# Prototype Orthopedic Bone Plates 3D Printed by Laser Melting Deposition

**DOI:** 10.3390/ma12060906

**Published:** 2019-03-19

**Authors:** Diana Chioibasu, Alexandru Achim, Camelia Popescu, George E. Stan, Iuliana Pasuk, Monica Enculescu, Stefana Iosub, Liviu Duta, Andrei Popescu

**Affiliations:** 1Center for Advanced Laser Technologies—CETAL, National Institute for Lasers, Plasma and Radiation Physics, 077125 Magurele, Ilfov, Romania; diana.chioibasu@inflpr.ro (D.C.); alexandru.achim@inflpr.ro (A.A.); camelia.popescu@inflpr.ro (C.P.); stefana.iosub@inflpr.ro (S.I.); 2Faculty of Applied Sciences, Department of Physics University Politehnica of Bucharest, 060042 Bucharest, Romania; 3National Institute of Materials Physics, 077125 Magurele, Ilfov, Romania; george_stan@infim.ro (G.E.S.); iuliana.pasuk@infim.ro (I.P.); mdatcu@infim.ro (M.E.); 4Lasers Department, National Institute for Lasers, Plasma and Radiation Physics, 077125 Magurele, Ilfov, Romania

**Keywords:** Ti6Al4V bone plates, laser melting deposition, titanium alloys, CAD-CAM design, in vitro testing

## Abstract

Laser melting deposition is a 3D printing method usually studied for the manufacturing of machine parts in the industry. However, for the medical sector, although feasible, applications and actual products taking advantage of this technique are only scarcely reported. Therefore, in this study, Ti6Al4V orthopedic implants in the form of plates were 3D printed by laser melting deposition. Tuning of the laser power, scanning speed and powder feed rate was conducted, in order to obtain a continuous deposition after a single laser pass and to diminish unwanted blown powder, stuck in the vicinity of the printed elements. The fabrication of bone plates is presented in detail, putting emphasis on the scanning direction, which had a decisive role in the 3D printing resolution. The printed material was investigated by optical microscopy and was found to be dense, with no visible pores or cracks. The metallographic investigations and X-ray diffraction data exposed an unusual biphasic α+β structure. The energy dispersive X-ray spectroscopy revealed a composition very similar to the one of the starting powder material. The mapping of the surface showed a uniform distribution of elements, with no segregations or areas with deficient elemental distribution. The in vitro tests performed on the 3D printed Ti6Al4V samples in osteoblast-like cell cultures up to 7 days showed that the material deposited by laser melting is cytocompatible.

## 1. Introduction

Recently, the additive technologies made the step from scientific research towards industry for the manufacturing of prototypes or niche parts with special shapes [1]. Even though higher production costs are envisaged for the mass production of parts using these technologies instead of conventional ones, the prospective advantages of 3D printing are hard to ignore [2]. Some worth mentioning benefits are as follows: (i)the lower manufacturing cost of prototypes;(ii)decreased manufacturing time by elimination of production steps;(iii)the possibility of manufacturing parts with composition gradients;(iv)design of parts with complex geometries, hard to obtain by other techniques;(v)decreased waste material amounts during parts production;(vi)costs reduction for equipment necessary for parts manufacturing. 

For the melting of metallic powders and shaping of 3D objects, three energy sources are typically considered: laser beams, electron beams, and spark plasma. 

For the particular case of metallic materials and using the aforementioned energy sources, there are two main approaches for creating 3D objects: directed energy deposition and powder bed fusion. For the laser-based additive manufacturing, two main printing techniques have been derived from these principles: Laser Melting Deposition-LMD [3,4,5,6,7,8] and Selective Laser Melting-SLM [9,10]. A comparison between the main features of interest for these two techniques is given in Table 1.

The aforementioned metal printing techniques are complementary since they cannot cover the whole range of 3D printing needs in the industry. LMD does not allow for high printing resolutions attained by SLM, but it permits the manufacturing of very large parts or completion with constructive elements of parts, impossible to be achieved by SLM [11]. 

Implantable medical devices represent a logical application target for 3D printing techniques due to the need for implants with customized shapes and dimensions, imposed by characteristics of the impaired tissue, envisaged to be treated, reinforced or replaced [12,13,14,15]. 

Ti6Al4V (also known as Ti grade 5) is a titanium alloy widely employed for the manufacturing of hip, knee, maxillo-facial or spinal endo-prostheses, dental fixtures or any other device to be used in intimate contact with bone, due to its good biocompatibility, low weight, excellent resistance to corrosion and high strength [16,17,18].

In this study, the LMD technique was chosen due to its good promise for the future development of large size implants, while minimizing the raw materials consumption. Furthermore, LMD is versatile, allowing its application for printing, coating, and alloying, and thus, opening the path towards innovation in the biomedical field. By LMD, Ti6Al4V composites can be synthesized in situ in combination with hard ceramic nanoparticles, such as TiC, in view of increasing its mechanical properties [19,20]. There are numerous papers reporting LMD experiments starting from Ti6Al4V powders, especially for cladding or repairing of mechanical parts [21,22,23]. While SLM is the technique of choice for producing additive manufactured medical devices (such as hip or knee prostheses, dental implants, cranial meshes or plates for facial reconstruction [24]), the LMD literature is limited to physicochemical tests of LMD deposited biocompatible alloys [25,26,27,28]. 

In a step forward, we report and discuss on the LMD manufacturing conditions leading to the fabrication of orthopedic bone plates. The main objective of the study was to produce and characterize Ti6Al4V shapes aimed for the future development of patient-customized 3D implants. The encountered technical difficulties that could impede this method to evolve from the research stage to the actual production of functional medical devices are underlined, and some possible technical solutions meant to overcome these shortcomings are highlighted. Answers to two highly relevant topical questions are provided: “Why 3D printing?” and “Why so many technological steps need to be involved in the implant manufacturing by 3D printing?”. Besides the optimization and the technological steps of the manufacturing process, a thorough structural and compositional investigation of the LMD-grown bulk material is presented together with cytocompatibility studies, assessing its preliminary biological performance. 

## 2. Materials and Methods

### 2.1. 3D Printing of Bone Plates

The Ti6Al4V bone plates were produced from a micro-sized powder purchased from LPW Technology, Widnes, UK. The experimental set-up is presented in Figure 1. For powder delivery, an automated powder feeder (Trumpf, Ditzingen, Germany) was used, which was connected through hoses to a nozzle with 3 flow channels that was mounted onto a robotic arm. For powder transport, a carrier gas was used, made of a He-Ar mix (1:3 ratio), with a flow rate of 11 L/min. After an optimization step, the optimal powder flow necessary for obtaining a dense deposited material and a high growth rate was found to be 3 g/min. Over 3 g/min, the increase in the growth rate was not significant enough to justify the high amount of consumed powder. Moreover, when surpassing 3 g/min of blown powder, an unwanted deposition of sputtered material occurred. This was caused by unmolten blown powder which adhered to the hot substrate in the vicinity of the laser irradiated area. Below 3 g/min, the quality of samples varied, from a discontinuous deposition (up to 1 g/min) to porous and rough samples (up to 2.5 g/min). 

The laser beam used for melting the powder was generated by a Yb:YAG disk source (λ = 1030 nm, continuous wave), model TruDisk 3001 (Trumpf, Ditzingen, Germany). The selected peak power after optimizations was of 1 kW. The beam was Gaussian, with a surface focused spot size of 600 µm diameter. The molten pool and the powder melting were monitored in real time using a camera attached to the robotic arm. 

The Ti substrates were not heated before LMD printing. However, the substrate temperature was obtained in situ during the LMD process with a chromel-alumel thermocouple. In our experimental conditions, during bone plates printing, the substrate temperature reached a maximum of 300 °C.

The robot was a Kr30HA model (Kuka Robotics, Augsburg, Germany), with 6 movement axes. The translation speed was set at 1 m/min, in order to obtain the best compromise between the thickness of traced lines, the coherence of printed material and deposition height. 

The bone plates were designed in the graphical engineering software SolidWorks^®^ (Dassault Systemes, Vélizy-Villacoublay, France) and subsequently imported in TruTops Cell^®^ (Trumpf, Ditzingen, Germany), a robot movement code generator. 

### 2.2. Machining Steps for the Manufacturing of the Final Shape

First, an incipient shape was printed and further sliced using a disk cutting machine, model Brillant 200 (ATM, Mammelzen, Germany). Intermediate bone plate shapes with the thickness of 1 mm were thus obtained.

The intermediate shape was drilled in order to obtain 4.5 mm diameter holes for the fixing screws, using a manual drilling machine MFB 16 Vario (Elmag, Tumeltsham, Austria). A spiral drill DIN 338 type VA HSSEC8 of 4.5 mm in diameter, oriented perpendicularly on the bone plate surface, was used for piercing. The rotation speed of the drill was of 450 rpm. Drilling was conducted using a lubricant (Premium Opta Cool 700S, Brehmen, Germany). The final manufacturing step was surface polishing. This step was conducted for 120 min using a vibro-finishing FKS06 machine (Rosler, Prescot, UK), with RSG6/6 polishing chips and an FC120 polishing compound. 

### 2.3. Physicochemical Characterization

The powder morphology was studied by scanning electron microscopy (SEM), using an EVO 50XVP (Carl Zeiss, Oberkochen, Germany) instrument. The particle size distribution analysis was performed on the basis of four SEM micrographs (collected on randomly selected regions with areas of 1340 × 905 µm^2^) with the help of Image J software (National Institutes of Health, MD, USA). A total of 697 individual particles were identified and measured.

The compositions of the source powder and LMD sample were assessed by energy dispersive X-ray spectroscopy (EDXS) by means of a 133 eV XFlash 4010 (Bruker AXS, Karlsruhe, Germany) attached to the SEM microscope. The EDXS analyses were performed in four different large (533 × 360 µm^2^) regions of the specimens in order to average over possible compositional non-homogeneities. Mean ± standard deviation values were computed. Further statistical analyses were carried out using the unpaired Student’s *t*-test, with differences being considered significant at a probability value (*p*) < 0.05. The surface homogeneity of the LMD sample was probed by EDXS elemental mapping.

The solid structures produced by LMD were cut in coupons for analysis, using a Brillant 200 machine (ATM, Germany), with rotating speed of the cutting disk of 2850 rpm. Sections were performed in four different locations of each LMD analyzed sample, in order to check for the results repeatability. The cut coupons were embedded in Bakelite by hot pressing (P = 50 bar, T = 150 °C) using Opal 410 (ATM, Germany) pressing machine with a 30 mm cylinder. The coupons were prepared for metallographic analysis by polishing to a mirror-like finish, using a Saphir 520 (ATM, Germany) polishing machine, equipped with an automatized head for fixation and translation of embedded samples.

In order to reveal the metallographic structure, the polished surfaces were chemically etched with a mix of HF (20%), HNO_3_ (10%), and water (70%). The samples treated with the chemical reagent were studied by optical microscopy using a DM4000 B LED instrument (Leica, Wetzlar, Germany). In order to evidence the compactness of the LMD synthesized material, samples unexposed to the reagent were also studied by optical microscopy. 

The hardness of the bulk structures was determined by a Vickers microdurimeter, model FM-700 (Future Tech, Holbrook, USA), using a load of 5 × 10^−2^ N. 

X-ray diffraction (XRD) investigations were performed with a D8 Advance (Bruker AXS Karlsruhe, Germany) apparatus using CuK_α_ (λ = 1.5418 Å) radiation and a rapid LynxEye^TM^ detector. The samples were scanned in symmetric geometry (θ–θ), in the 2θ angular range 25°–75°, using a step of 0.04° and time per step of 2 s. The XRD analysis aimed to provide qualitative and quantitative information about the crystalline phases. The XRD data processing was performed using the TOPAS^®^ (Bruker, Karlsruhe, Germany) software. A corundum reference sample (NIST SRM 1976) was used to verify the alignment of the instrument and to model the instrumental line profiles.

### 2.4. In Vitro Testing

LMD printed samples were cut in 3 × 6 mm^2^ coupons for in vitro testing. Before the biological assays, all the mirror-polished coupons were steam-sterilized at 121 °C for 30 min, using an AES-8 autoclave (Raypa, Barcelona, Spain).

#### 2.4.1. Cell Culture

Human osteosarcoma cells (SaOs2) were cultured in McCoy’s 5A medium (Gibco, Waltham, USA) supplemented with 15% fetal bovine serum (FBS) (Gibco, Waltham, MA, USA) and 1% penicillin (10,000 U/mL)-streptomycin (10000 µg/mL) (PEN-STREP) (Gibco, Waltham, MA, USA). The cells were split at passage P19 and seeded on the coupons placed in 24-well tissue-culture test plates (TPP Techno Plastic Products AG, Trasadingen, Switzerland) at a density of 15000 cells/sample. They were cultured in a 5% CO_2_ humid atmosphere at 37 °C for 1, 3, and 7 days. Glass coverslip (CS) of 12 mm diameter was used as an experimental control. The cells cultivated on CS were routinely visualized by using a Leica DMi1 (Leica Microsystems, Wetzlar, Germany) inverted phase contrast microscope. 

#### 2.4.2. MTS Assay

Cell proliferation was investigated using a CellTiter 96 aqueous solution cell viability kit from Promega (Madison, WI, USA). The assay is based on the use of a tetrazolium compound (3-(4,5-dimethylthiazol-2-yl)-5-(3-carboxymethoxyphenyl)-2-(4-sulfophenyl)-2*H*-tetrazolium, inner salt—MTS) that is chemically reduced by viable cells into formazan, which is soluble in the tissue culture medium. Since the production of formazan is proportional to the number of living cells, the intensity of the produced color can be used as an indicator of cell proliferation.

The samples were seeded in duplicates for 1, 3, and 7 days. At each time interval, the samples were transferred to new clean well plates and subsequently incubated at 37 °C for 1 h and 30 min with fresh media containing MTS reagent. Afterward, the absorbance values were recorded at 450 nm on an LB 913 Apollo 11 spectrophotometer (Berthold Technologies, Bad Wildbad, Germany). The results were subtracted to background represented by MTS mix with culture media in the absence of cells and represented as a mean ± standard deviation.

#### 2.4.3. Immunofluorescence Microscopy

Cells grown on all samples were examined by means of fluorescence imaging. After 24 h and 72 h of culture, SaOs2 cells were fixed with 4% paraformaldehyde at room temperature and kept in phosphate-buffered saline (PBS) at 4 °C before labeling. The fixed cells were then permeabilized with 0.2% TritonX-100 and blocked in 0.5% bovine serum albumin. In order to visualize the actin filaments, the cells were stained with Alexa Fluor 488-conjugated Phalloidin (Cell Signaling, Technology, Danvers, MA, USA). The cells were then treated with 1 μg/mL Hoechst (Cell Signaling, Technology, Danvers, MA, USA) in order to label the nuclei. After each incubation, the samples were washed three times with PBS. In the end, the specimens were analyzed on glass slides using a DM 4000 B LED fluorescence microscope equipped with a DFC 450 C camera (Leica Microsystems, Wetzlar, Germany) with appropriate filters.

#### 2.4.4. SaOs2 Cell Morphology

The SaOs2 cell morphology was studied by SEM. For SEM studies, the cells were fixed after 1 and 3 days of interaction with samples. For fixation, 2.5% glutaraldehyde was used for 45 min at room temperature and washed two times with PBS. Samples were kept in PBS until the process of dehydration. This procedure involved successive immersion in 70%, 90%, and 100% ethanol (EtOH), 15 min twice for each concentration. Cells were then incubated by sequential incubation in 50%:50%, 25%:75%, and 0%:100% solutions of EtOH:hexamethyldisilazane (HMDS), two times for 3 min in each combination. The specimens were dried and metalized prior to the microscopy investigations. The metallization consisted of a 10 nm thin gold layer deposited by magnetron sputtering on the surface of specimens with a manual sputter coater (Agar Scientific, Essex, UK).

## 3. Results

### 3.1. Powder Characterization

Representative SEM images of the Ti6Al4V source powder, together with the particle size distribution histograms, are presented in Figure 2. The powder particles are spherically-shaped (Figure 2a). At higher magnification (Figure 2b), small spherical particles soldered to the prominent larger particles, both having a smooth morphology. The particle size analysis, performed on the basis of SEM micrographs, revealed two particle populations (Figure 2c): (i) one with diameters in the range 10–40 µm, best-approximated with a log-normal distribution having a median value of ~12 µm and (ii) one with sizes in the range 50–130 µm, well-fitted by a Gaussian function, with a median value of ~71 µm. The composition of the Ti6Al4V powders was determined by EDXS measurements. A representative EDXS spectrum is shown in Figure 2d, which highlights the high degree of the powder purity (at the sensitivity limit of the analysis technique), with only Ti, Al, and V peaks being evidenced. 

### 3.2. LMD Optimizations

The first printing tests were conducted to identify the optimal (*i*) laser power and (*ii*) scanning speed for melting the blown powder. A 150 mm linear trajectory of the robot arm was programmed, and the material was deposited during two consecutive passes, varying the scanning speed between 0.3 m/min–1 m/min. 

Experiments were started with a laser power of 300 W that proved insufficient for depositing the material. The powder was molten starting with 800 W peak power, but the deposition was discontinuous and inefficient (Figure 3a). For a peak power of 1000 W, a continuous, clearly defined line of 2 mm thickness was produced (Figure 3b). Above 1000 W, the deposition became rough, the laser plasma was very intense, and the molten material was sputtered in the vicinity of the irradiation site (Figure 3c). Such structures were not conforming to our requirements, as the resolution of 3D samples to be produced would have been coarse. Another testing variant was to increase the scanning speed by three times while keeping the laser power constant at 1200 W. Figure 3d presents a line traced in the aforementioned conditions. The increase in quality was evident, and the number of particles around the irradiated area was drastically decreased. However, at the end of the LMD experiment, the unmolten powder found in the glass chamber was visibly higher with respect to the experiment conducted at 1000 W and scanning speed of 1 m/min. This could translate into hundreds of grams of powder material being wasted during the printing of complex shapes that require hours-long manufacturing irradiation durations. After analyzing all structures, the optimal process parameters for printing the walls (to be used as specimens for the physicochemical characterizations) and 3D shapes were chosen to be: peak power of 1000 W and scanning speed of 1 m/min. These selected parameters allow for a higher yield of the processing because the scanning speed is slower and more powder is blown in the molten area, while the thickness of the printed line is 1/3 thinner as compared to the case of 1200 W irradiation, which warrants a better resolution of 3D shapes. 

In these conditions, walls of Ti6Al4V similar to the one depicted in Figure 4 were grown by LMD by moving the robot along a straight line. Each built layer was 0.2 mm thick and the nozzle was raised accordingly after each pass. The walls were then cut into coupons and prepared for physicochemical characterizations. 

### 3.3. Morphological and Structural Characterizations of LMD Grown Structures

Figure 5a displays a characteristic optical microscopy image of a mirror-polished LMD grown structure. The printed material was homogenous and free of pores or cracks. This was consistent for the whole bulk of five studied samples. Four cuts were performed in random places for each bulk in order to check for porosity or other defects. Figure 5b presents an optical microscopy image of a Ti6Al4V coupon surface after the metallographic attack. Polyhedral grains of 200–400 µm in length were clearly visible. These are the primary β-phase grains, formed when the deposited material surpassed 882 °C, that is, the crystallographic temperature’s transformation from α-Ti (hexagonal close-packed) to β-Ti (body-centered cubic). The printed material contains a mix of α- and β-phases. The α-phase, light colored in Figure 5b,c, is spread through the β-phase (appearing dark colored in Figure 5c). 

For a more detailed investigation of the crystalline phases and for quantitative assessments, the powder and the printed LMD material were investigated by XRD. The XRD patterns are displayed in Figure 6.

The XRD patterns were fitted using the Pawley method [29] aiming to estimate the lattice parameters from the line positions and the crystallite sizes from the line breadths. For both the powder and LMD material, the line profiles could only be simulated if we consider both small size and microstrain line broadening. This indicated that microstrain was large and could not be neglected. Microstrain measures the fluctuations of the interplanar spacings (d) of a crystalline phase relative to the corresponding mean value occurring in the investigated sample volume, ε_0_ = <ε^2^>^1/2^, where ε = Δd/d and <…> indicates the averaging operation. This structure parameter quantifies the local disorder of the lattice and may have different sources, such as lattice deformations, due to mismatching substitutional atoms or interstitial atoms, compositional inhomogeneity or non-relaxed crystallite boundaries [30].

The powder consisted of a unique Ti-Al-V phase with hexagonal close-packed (hcp) crystal structure (space group P63/mmc), which corresponds to the low-temperature α-phase of Ti-Al-V [31,32]. We used the diffraction file ICDD-PDF4+: 04-002-8708, chemical formula Ti_90_A*l*_6_V_4_, as a reference for α-phase which corresponds to the grade Ti-6Al-4V [33]. The quality of the fit is shown in Figure 7a. The experimental lattice constants were a = 2.924 Å and c = 4.659 Å, and the unit cell volume of 34.79 Å^3^ was found smaller than the unit cell of the above-mentioned reference phase (34.84 Å^3^) and larger than the unit cell of hexagonal structure of a Ti_94_Al_5_V_1_ alloy (34.38 Å^3^) (ICDD-PDF4+: 04-020-7055 [34]). This is probably related to small deviations from the nominal composition. The average size of the crystallites was found to be around ~50 nm. A large microstrain, ε_0_ = 6%, was obtained.

The XRD investigations of the solid structure printed by LMD revealed, besides the preponderant hcp α-phase, the presence of remnant high-temperature β-Ti-Al-V phase with body-centered cubic structure (bcc, space group Im–3m) [29,31]. Both the α- and β-phases show deviations of the relative intensities from those of the standard structures given in the certified XRD powder diffraction files. We used the above-mentioned file (i.e., ICDD-PDF4+: 04-002-8708) for the α-phase and the ICDD-PDF4+: 04-018-5433 file, chemical formula Ti_76_Al_6_V_18_, for the bcc β-phase [35]. This phase had the closest lattice constant (a = 3.216 Å) with respect to the one determined experimentally for our bcc structure (a = 3.204 Å). The deviations of the line intensities are related to some preferred orientations of crystallites in both phases (rather than to compositional deviations from the reference phases). This seems a bit more pronounced in the β-phase. The Pawley whole powder pattern fitting method overlooks the line intensity variations due to texturing, thus it is very useful in determining lattice constants and crystallite sizes for textured phases. The fit is displayed in Figure 7b. The lattice parameters of the α-phase were of a = 2.932 Å, c = 4.701 Å, with a unit cell volume of 35.00 Å^3^. The mean crystallite size of the α-phase was found to be around 80 nm, larger than for the source powder, while the microstrain was smaller, ε_0_ = 2%, suggesting that the lattice is on average more relaxed. The crystallites of the β-phase were much larger, 200 nm, but with a large degree of local deformation or inhomogeneity (microstrain), ε_0_ = 8%. The “disorder” of the β-phase is probably related to the non-equilibrium nature of this phase at room temperature, this phase being preserved at cooling due to mechanical constraints caused by the α-phase expansion. An attempt has been made to determine the weights of α- and β-phases in the LMD sample, by fitting the XRD diagram with the Rietveld method [36] (Figure 7c). This procedure is based on fitting the powder diffractogram with a curve simulated on the basis of the atomic structures of the unit cells of each crystalline phase. The scale factor of each phase is determined by the intensities of the diffraction lines of that phase, which in turn is related to the number of unit cells in the investigated volume, which is proportional with the volume and mass of that phase. Thereby, weight percentages can be calculated. The resulting mass concentration of the β-phase was around 30%. As can be observed in Figure 7c, the line intensities are not perfectly fitted neither for α- nor for β-phase, in spite of using different texture models, available in the fitting software. Thus, this weight percent should be only considered as approximate

Micro-hardness tests were conducted on mirror polished coupons cut from LMD printed bulks in 30 randomly selected points. The mean ± standard deviation value of hardness was of 392 ± 7 HV. 

EDXS quantitative compositional analyses were conducted, both on the source powder and LMD printed samples. Figure 8 displays the comparative elemental concentrations (wt.%). In the case of Al and V, the differences in composition were not statistically significant (*p*-values of 0.65288 and 0.23048 for Al and V, respectively). However, for Ti, a slight drop in weight concentration by 0.92% was recorded after printing, which was found statistically significant (Student’s *t*-test, *p* = 0.03748). Still, the difference in Ti is close to the resolution limit of EDXS method, situated according to International Organization for Standardization (ISO) standard 2309:2011 (“Microbeam analysis—Quantitative analysis using energy-dispersive spectrometry (EDS) for elements with an atomic number of 11 (Na) or above”) at one wt.% for elements with atomic number Z > 10.

Figure 9 presents the elemental distribution, on a randomly selected sample region, as evidenced by EDXS mapping. There was no visible elemental segregation or areas with poor elemental distribution for either Ti, Al or V. All elements were homogeneously distributed over the investigated area, confirming the biphasic α + β solid solution structures. 

### 3.4. Bone Plates Manufacturing

After identifying the printing regime that allows for deposition of metal structures with a composition close to that of the source powder, uniform elemental distribution, and absence of porosity or cracks, the next step was to print a metal bone plate. The bone plate shapes were designed in SolidWorks^®^ and afterward imported in TruTops Cell^®^. Here, a meander was drawn over the bone plates surface with a distance between lines of 2 mm (Figure 10a). This was the path followed by the robotic arm during printing. The distance between the meander lines was chosen to be equal to the width of a printed line. To obtain the incipient bulk shapes from which the Ti6Al4V bone plates would be modeled, the robot traced 10 times consecutively on the meander trajectory. Between each trajectory, the robot moved 2 mm upwards on the z-axis. Figure 10b presents the meander trajectory multiplied 10 times, with a height of 2 mm between the contours. Ti disks of 100 mm diameter and 10 mm thickness were used as substrates. The incipient bulk bone plate shape, that is, the result of a 3D printing with 10 superposed layers, is presented in Figure 11.

From the first printing tests, it was obvious that the scanning direction played an important role in the aspect of the printed part. Therefore, three scanning possibilities were envisaged: transversal scanning (Figure 12a), longitudinal scanning (Figure 12b), and longitudinal scanning with a traced contour (Figure 12c).

Transversal scanning was a failure. The central part of the incipient form was correctly traced, but the areas close to the borders of the circular elements were molten (Figure 12a-right). Melting was caused by the meander lines shrinking close to the edges of the circular shapes, necessary for filling the incipient form area. The shrinking of the meander produced the scanning of a small area surface for a prolonged time interval. This generated an augmented local heat transfer which had, as a consequence, the melting of the printed form edges. 

Longitudinal scanning (Figure 12b) marked an evident improvement of the incipient form printing. By visual inspection, the form had apparently the required aspect for proceeding with the next production step, that is, cutting for obtaining an intermediate form with the thickness specified by the technical drawing. However, the edges of the intermediate form were corrugated, each protrusion representing a change of direction of the robotic arm on the meander trajectory (Figure 13a). In order to avoid a supplementary edge milling step, a printing variant of an incipient form was tested, with a longitudinal meander trajectory, followed by a supplemental contour tracing of the printed incipient form. The result is given in Figure 12c. 

The incipient form contour was, in this case, much more pronounced. This new incipient form was further cut, and the intermediate shape is shown in Figure 13b. The unwanted roundness of the incipient form edges, caused by liquid metal flow when printed with a meander trajectory without drawn contour, was greatly diminished. After cutting, the bone plate edges were significantly sharper as compared to the case of printing without specific contour drawing (Figure 13b vs. Figure 13a). 

The next step in the bone plate manufacturing was drilling of the intermediate shape, according to the technical drawing (Figure 14a). A photo taken during the drilling operation is given in Figure 14b.

The final production step was of polishing by vibro-finishing. The surface polishing is produced by the relative movement between ceramic polishing chips and the bone plates, caused by the vibrations produced by the finishing machine. The final form of some bone plates is presented in Figure 15. 

### 3.5. In Vitro Tests in Cell Cultures

In vitro studies in osteoblast-like cell cultures were conducted in order to evaluate the eventual cytotoxicity effects that the printed material can induce. Even though casted Ti6Al4V is commonly used as a biomaterial for implants and prostheses, there is a possibility that during LMD printing, the appearance of a particular microstructure or phases (i.e., the bcc β-phase) might occur. The potential incompatibility with cells adhesion and proliferation or even the advent of toxic effects induced by the presence of such compositional or morpho-structural features should not be disregarded. The in vitro assays removed any doubts concerning the cytotoxicity of the mirror-polished surface of LMD printed Ti6Al4V material.

The cells proliferation results, as obtained by the MTS assay, are displayed in Figure 16a. As a result of mitochondrial activity, cells degrade the MTS salts to formazan, which absorbs the 450 nm wavelength. Thus, a higher absorbance is indicative of the presence of a higher number of cells. In Figure 16a, one can observe an exponential trend for cells proliferation from day 1 to day 7. Three days after seeding, the absorbance values were ~10% higher as compared to day 1, while after 7 days, they were ~40% higher as compared to day 1. This constitutes a qualitative proof that the LMD printed Ti6Al4V samples are non-toxic for SaOs2 cells. 

Figure 16b presents the SEM morphology of SaOs2 cells grown onto the LMD printed Ti6Al4V. There are strong interactions between cells and printed material. The SaOs2 cells were well-spread and well-adhered, occupying a large area and generating cellular protrusions in order to maximize the contact with the LMD printed Ti6Al4V surface. Further, the fluorescence microscopy images showed the degree of cellular surface coverage at 1 (Figure 16c) and 3 (Figure 16d) days after seeding. After three days, the SaOs2 cells proliferated and spread on the whole surface of the LMD printed Ti6Al4V biomaterial. 

## 4. Discussion

The metallographic structure obtained under our deposition conditions is not common for LMD. Usually, LMD and SLM printed Ti6Al4V structures have a characteristic α’ martensitic hexagonal structure with lattice parameters of a = 2.93 Å and c = 4.67–4.68 Å [37]. However, in special conditions (i.e., by slowing down the scan speed during SLM, and thus enabling a long interaction time between the laser beam and the irradiated material), the formation of an α + β more stable structure is possible [38,39]. Generally, a very fine structure with α grains having lengthiness of 1–3 μm and thickness of 0.5–1 μm is obtained [38]. α + β structures are much more common for electron beam melting [40]. The microstructure similarity between our LMD printed structures and electron beam melted ones can be explained by the high build temperature met in both cases. The high temperature is intrinsic to the process of electron beam melting due to the chamber temperature built during processing (typically of 650–700 °C) [31], while, in our case, it is due to the very slow processing speed that allowed for heat accumulation. Thus, an in situ heat-treatment occurred, that induced the transformation of the α′ martensitic phase into the equilibrium α + β microstructure. 

The hardness values obtained by micro-indentation of LMD printed samples were slightly higher than the ones reported for casted and slowly cooled Ti6Al4V: 392 ± 7 HV vs. ~349 HV [12]. By comparison, R. Raju et al. [23] also obtained by LMD of Ti6Al4V an α + β microstructure and reported a micro-hardness of 360 ± 10 HV. Baufeld et al. [41] obtained by wire-based laser beam deposition an α + β microstructure with a hardness response of ≈332 HV. In fast cooling regimes, when β grains transform into martensitic α’, the expected hardness is higher. L. Song et al. [32] reported Ti6Al4V fabricated by direct laser cladding, a process very similar to LMD and devoted to the additive fixing of defective parts, and a micro-hardness value of ~550 ± 20 HV.

The difference in hardness between our samples and the casted ones could be caused by the formation of very fine α needles in β grains that created a basketweave-like reinforced structure. This microstructure can stop plane dislocations, and thus improve hardness. Mercelis and Kruth [42] attributed the increase of hardness to the high residual stresses caused by the rapid heating-cooling cycles, but this seems not to be the case in our low cooling regime.

Related to the bone plates manufacturing technology, one could wonder: “Why were so many technological steps involved in the implant manufacturing by 3D printing?” In the ideal case, the plates could be printed in a single step on a ceramic substrate, with the desired dimensions and holes. The ceramic would be broken at the end of the printing process, and the resulting implant would only require a surface finishing step.

Ti6Al4V is a challenging alloy to deposit by LMD. On one hand, it behaves well as it does not develop porosity or cracks due to fast heating-cooling cycles, but on the other hand, it is complicated to find a suitable substrate for deposition. Ti6Al4V has a coefficient of thermal expansion (CTE) of ~8.6 × 10^−6^ °C^−1^ [43], very different from that of steel or refractory ceramics. Therefore, liquid Ti6Al4V would be hydrophobic for such type of substrates, forming droplets and resulting in an uncontrollable printing process. The only matching substrates are based on Ti on which Ti6Al4V grows homogeneously. We were therefore limited to the use of Ti plates on which we grew our incipient shapes, which had to be further mechanically processed in order to reach the final forms. We are now currently investigating some carbonaceous substrates which have CTE values close to those of Ti6Al4V and delivered promising results during preliminary tests. 

Another question that a reader can legitimately address would be “Why 3D printing?” for such plates that can be easily cut from a metal sheet and further submitted to a surface finishing procedure. With classic cutting methods, productivity would also be much higher than in the case of 3D printing. There are two arguments in favor of 3D printing:

(i) The first rationale is related to the design. While a cutting machine can produce a metal plate, it cannot produce a complex design. For 3D printing, we presented just a simple trial showing that the technique is compatible with the implant design and fabrication. There are currently commercial software that generate between imposed limits, shapes with futuristic designs that maximize the mechanical resistance with minimal consumption of material. This can accommodate the future development of implants with lower mass and better mechanical properties. Additive techniques can accommodate the production of such complex shapes with much ease with respect to the subtractive methods.

(ii) The second argument derives from the composition tuning. There are endless possibilities of research in terms of in situ alloying, manufacturing of implants with gradient or blended composition, manufacturing of a core from a material surrounded by a different enveloping material or multilayer structures.

## 5. Conclusions

1. Orthopedic bone plates of Ti6Al4V with shape and dimensions similar to the commercial laser cut ones, starting from powder materials with two populations of spherical particles (with diameters in the ranges 10–40 µm and 50–130 µm), were printed by LMD.

2. By optimizing the laser and scanning parameters, dense depositions with no porosity or cracks and with excellent compositional uniformity were obtained. 

3. The design of the trajectory followed by the laser beam proved essential in obtaining shapes that respect the dimensions and aspect of the technical drawings. The incipient 3D shapes obtained by 3D printing were further engineered by cutting, drilling and polishing in order to obtain the final bone plates.

4. The metallographic analyses showed that the samples lack porosity or defects usually caused by rapid heating/cooling cycles. The metallographic attack revealed α+β grains with a cubic phase matrix in which hexagonal α grains proliferated. The micro-hardness of the LMD printed Ti6Al4V was higher than that of the casted material, probably due to a basket-like interweaved grain structure, finer than the standard Widmanstatten structure of a casted material. 

5. The LMD printed Ti6Al4V alloy was biocompatible. There were no surface features or compositional deviations that could cause a negative response from cells. The in vitro assays showed that the osteoblast-like cells survived and proliferated well in the following seven days after seeding.

Although this method for bone plate manufacturing is more laborious than the conventional laser/plasma cutting operation for obtaining orthopedic plates, it opens horizons for special applications hardly accessible by other techniques, such as implants of composite materials, implants with variable composition, in situ alloying or implant thickness control. Moreover, with the proper substrate, LMD could be a printing technique suitable for large-scale implant fabrication.

## Figures and Tables

**Figure 1 materials-12-00906-f001:**
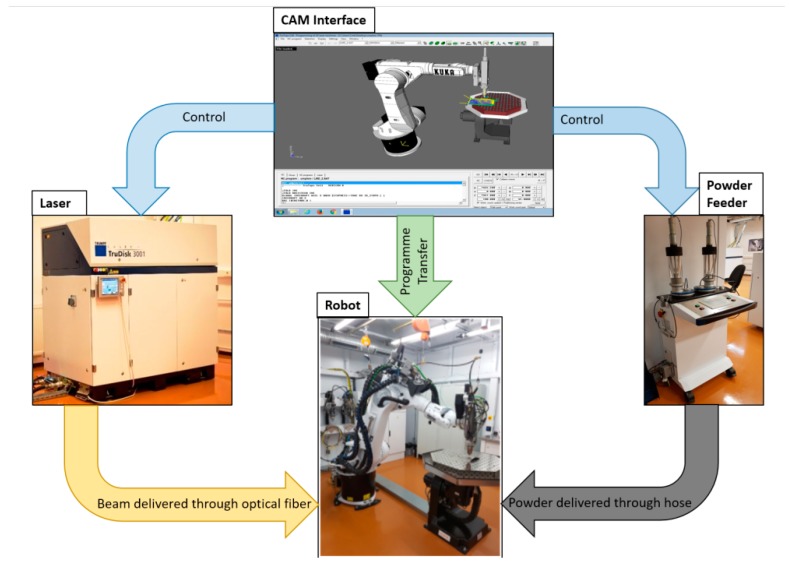
The experimental set-up used for the Laser Melting Deposition (LMD) experiments, comprising a laser source TruDisk 3001, a Kr30HA robot, and a powder feeder.

**Figure 2 materials-12-00906-f002:**
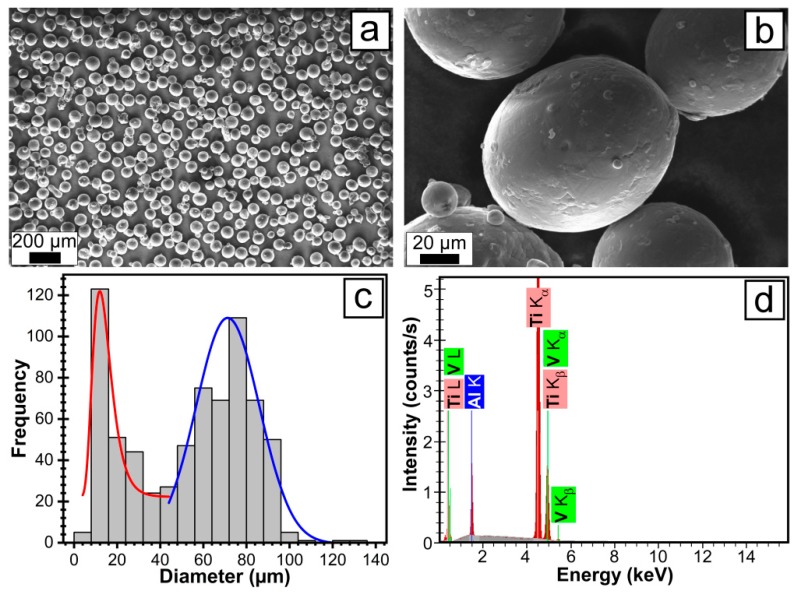
(**a**) Overview of the SEM micrograph for Ti6Al4V powder used as source material for LMD manufacturing of prototype bone plates; (**b**) Detailed SEM morphology of the small micron-sized particles adhering to the 50–130 µm diameter large particles; (**c**) Particle size distribution histograms; (**d**) EDXS spectrum characteristic of the Ti6Al4V powder.

**Figure 3 materials-12-00906-f003:**
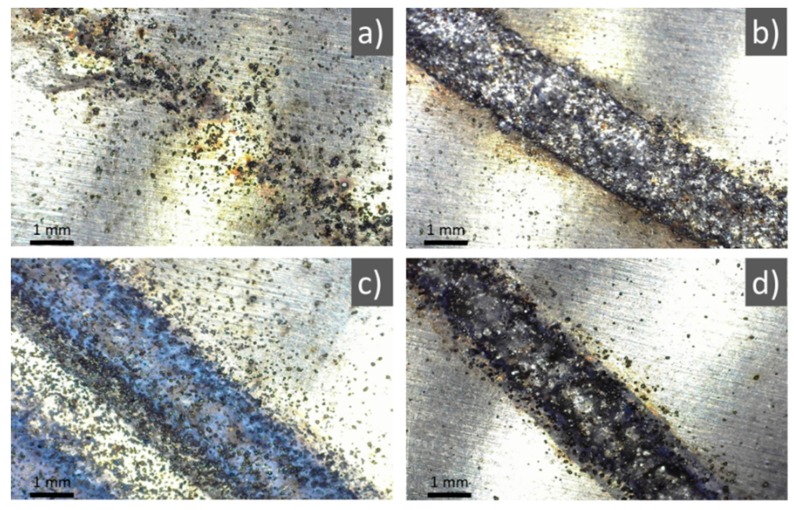
Ti6Al4V deposited along a linear trajectory during two passes of the laser beam and powder nozzle: (**a**) 800 W peak power and 0.3 m/min scanning speed; (**b**) 1000 W peak power and 0.3 m/min scanning speed; (**c**) 1200 W peak power and 0.3 m/min scanning speed; (**d**) 1200 W peak power and 1 m/min scanning speed. Magnification bar: 1 mm.

**Figure 4 materials-12-00906-f004:**
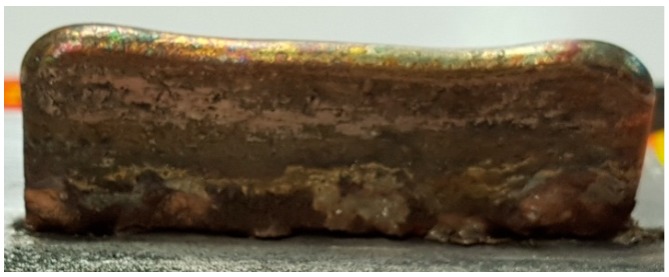
Ti6Al4V wall grown by Laser Melting Deposition (LMD).

**Figure 5 materials-12-00906-f005:**
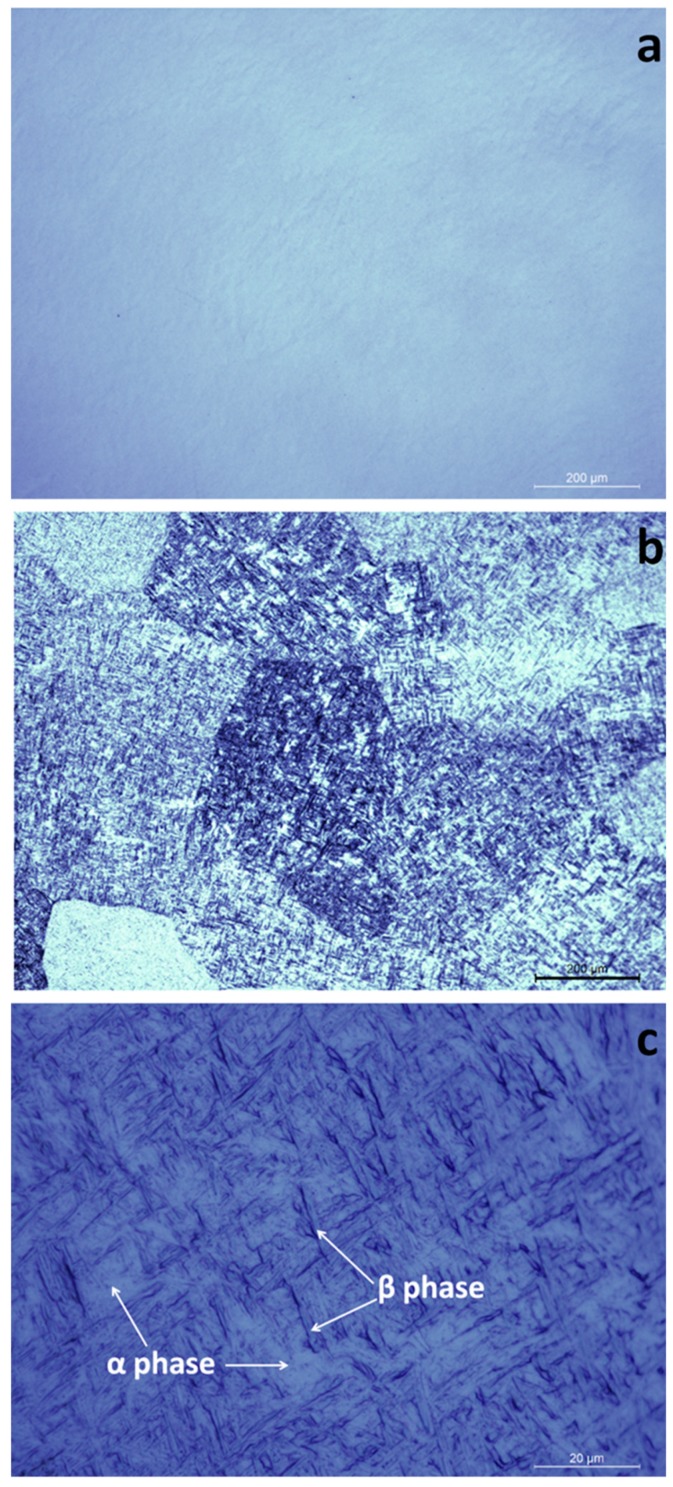
Surface optical microscopy images of (**a**) polished Laser Melting Deposition (LMD) grown sample and (**b**,**c**) LMD sample after the metallographic attack. Magnifications: (**a**,**b**) 10x, (**c**) 100x.

**Figure 6 materials-12-00906-f006:**
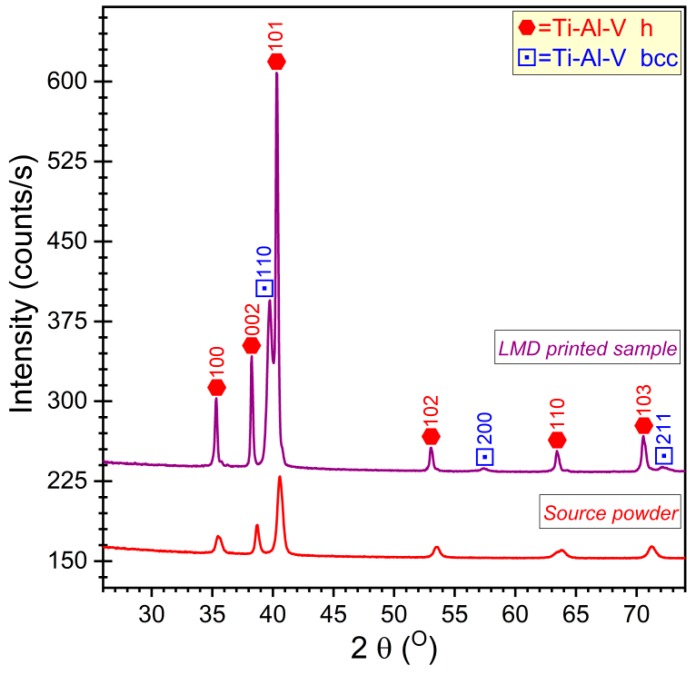
Comparative XRD patterns of the Ti6Al4V source powder and Laser Melting Deposition (LMD) printed bulk material.

**Figure 7 materials-12-00906-f007:**
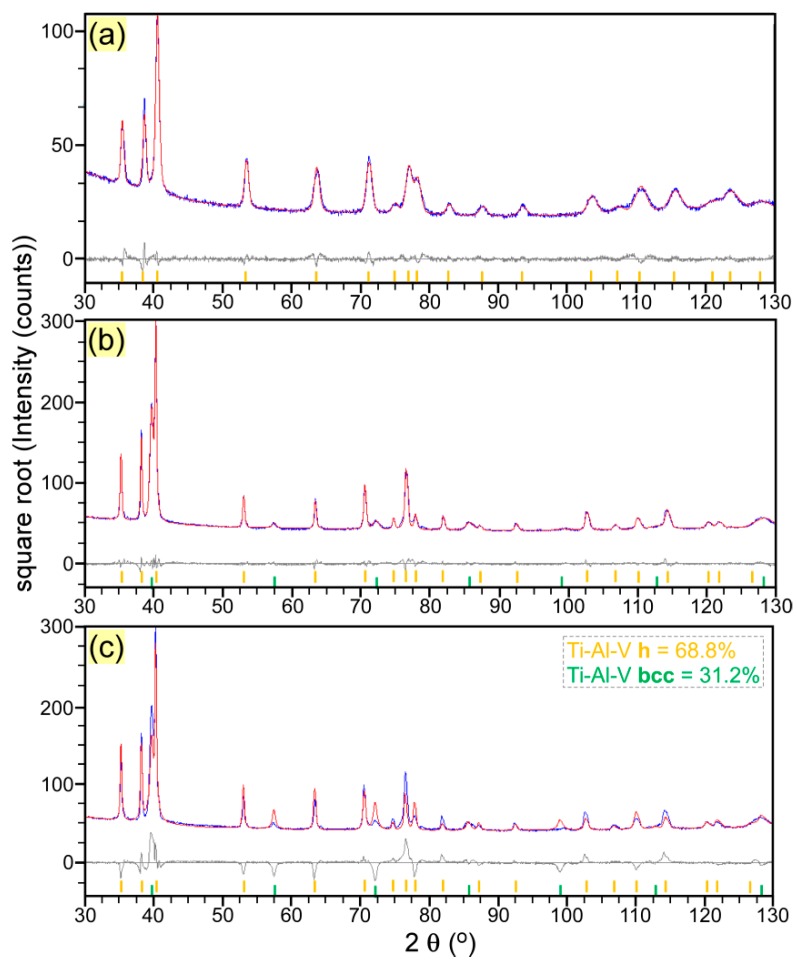
Experimental (blue) and simulated (red) XRD patterns and difference curves (gray) of the source powder fitted with Pawley method (**a**) and LMD printed Ti6Al4V fitted with Pawley (**b**) and Rietveld (**c**) methods. Orange sticks = α-phase; Green sticks = β-phase.

**Figure 8 materials-12-00906-f008:**
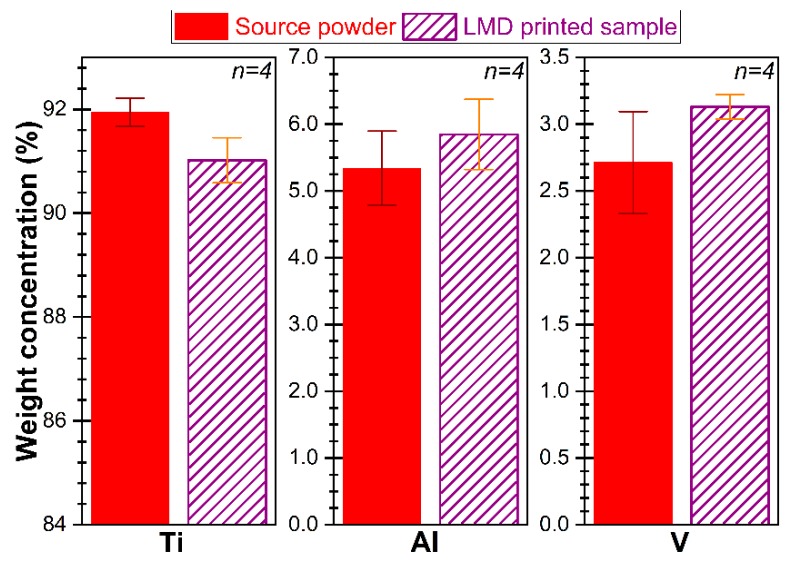
Comparative concentrations for Ti, Al, and V estimated by EDXS for the source powder and solid structure printed by LMD.

**Figure 9 materials-12-00906-f009:**
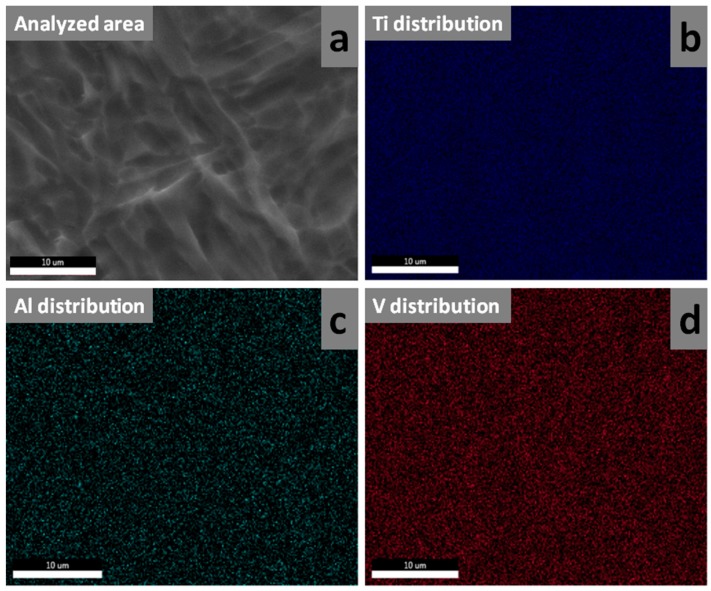
(**a**) SEM microscopic field selected for EDXS mapping. Elemental distribution maps for (**b**) Ti, (**c**) Al, and (**d**) V. Magnification bar: 10 μm.

**Figure 10 materials-12-00906-f010:**
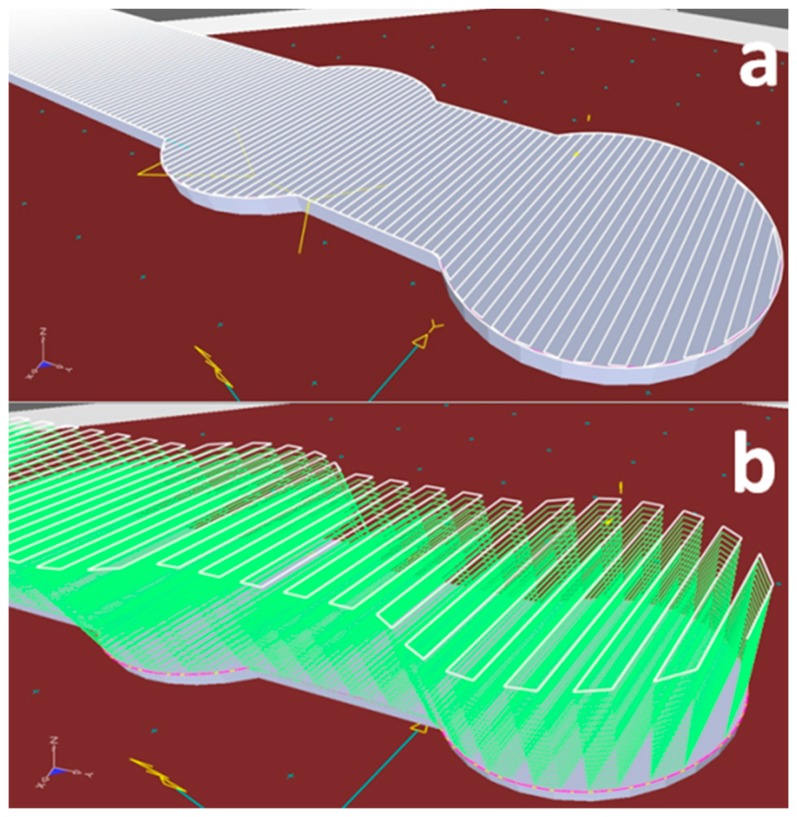
(**a**) Bone plate 3D drawing designed in SolidWorks^®^ and subsequently imported in TruTops Cell^®^. A robot path in the form of a meander follows the surface and contour of the bone plate; (**b**) Superposed meanders used for printing a solid incipient bone plate shape.

**Figure 11 materials-12-00906-f011:**
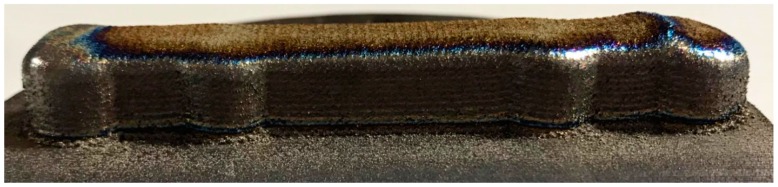
Incipient bulk bone plate shape obtained by Laser Melting Deposition (LMD) printing on a Ti substrate.

**Figure 12 materials-12-00906-f012:**
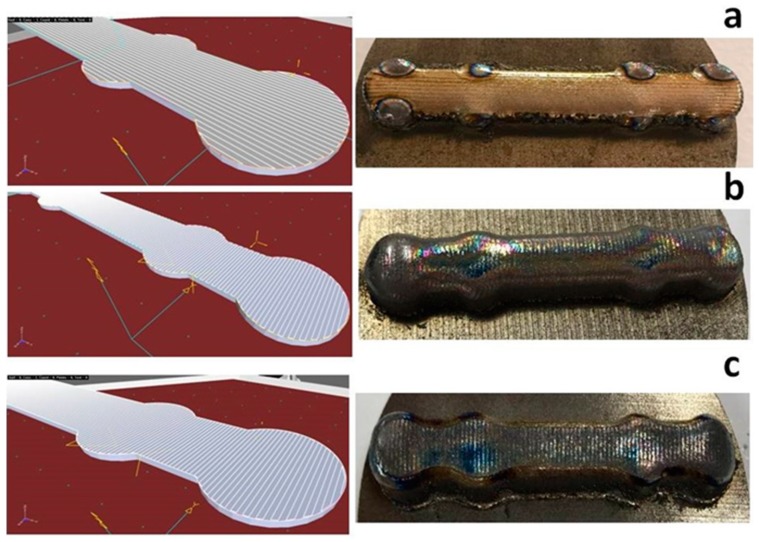
Scanning mode during printing and the resulting incipient form: (**a**) transversal scanning, (**b**) longitudinal scanning, and (**c**) longitudinal scanning with the supplemental contour.

**Figure 13 materials-12-00906-f013:**
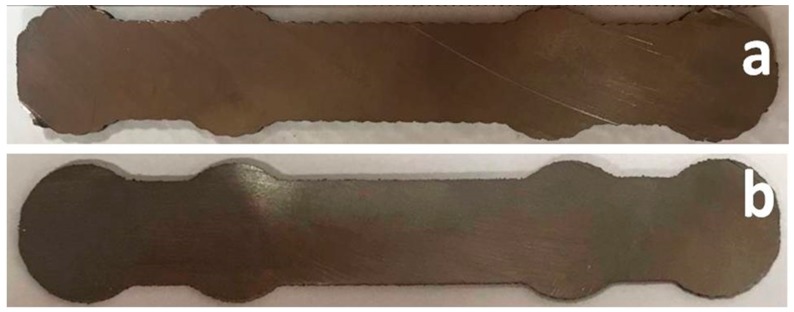
(**a**) Intermediate form with waved edges, obtained by Laser Melting Deposition (LMD) printing following a longitudinal meander trajectory and (**b**) intermediate form with improved edges by supplemental tracing of contour.

**Figure 14 materials-12-00906-f014:**
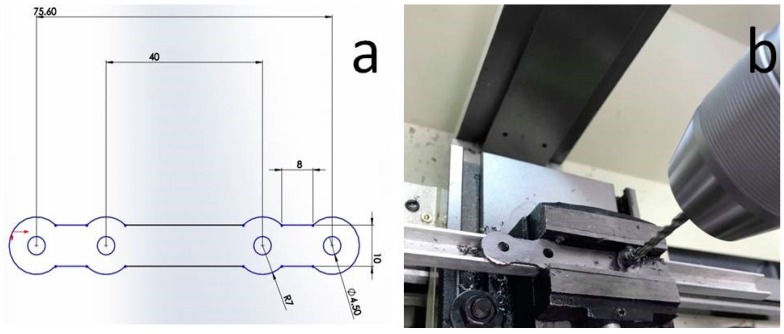
(**a**) Technical drawing of the bone plate and (**b**) intermediate shape drilling, according to technical drawing specifications.

**Figure 15 materials-12-00906-f015:**
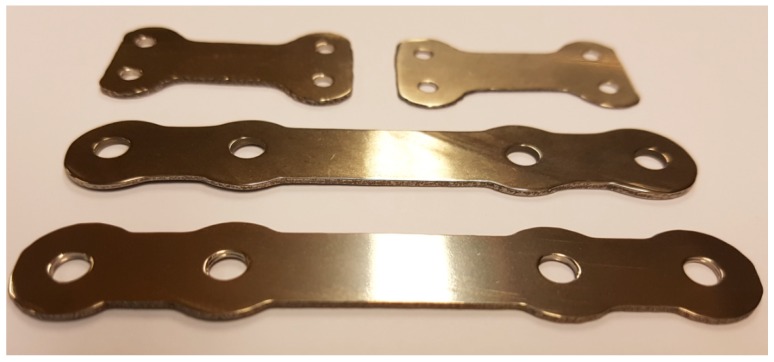
Photo of bone plates in the final form produced by Laser Melting Deposition (LMD) and mechanically processed for dimensions and roughness requirements.

**Figure 16 materials-12-00906-f016:**
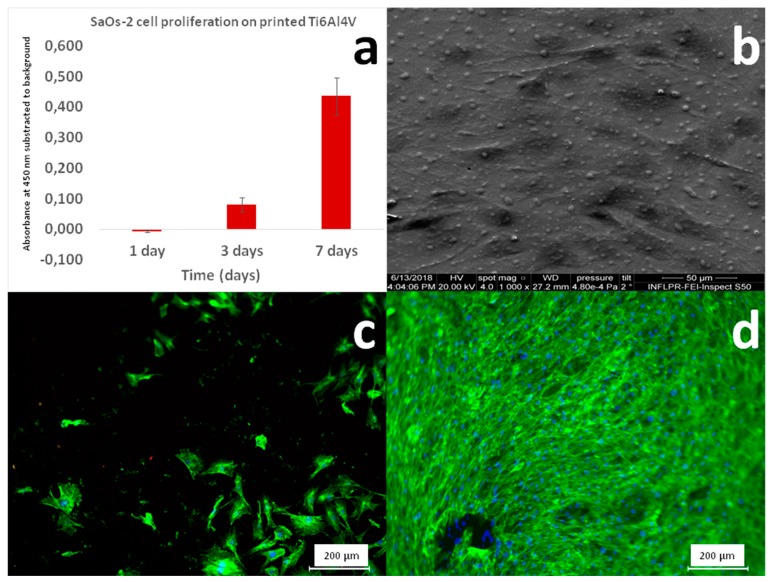
(**a**) MTS results for the SaOs2 cells grown on Laser Melting Deposition (LMD) printed Ti6Al4V; (**b**) SEM micrograph of SaOs2 cells 3 days after seeding on LMD printed Ti6Al4V; Fluorescence microscopy images of SaOs2 cells cultivated for (**c**) 1 day and (**d**) 3 days on the surface of LMD printed Ti6Al4V.

**Table 1 materials-12-00906-t001:** Comparison between various features of interest for LMD and SLM.

Feature	Laser Melting Deposition	Selective Laser Melting
Raw materials	Powder, wire	Powder
Heat source	Laser	Laser, electron beam
Technology	Powder is sprayed through a nozzle and melted by a laser beam	Beam transfers heat that melts a powder bed
Typical materials	Metals, ceramics	Metals, ceramics, polymers
Limitations by direction/axis	No	Yes
Resolution	Low	High
Versatility	High: used for coating, parts manufacturing, and in situ alloying	Low: limited to parts production
Parts size	Usually large scale objects	Usually small scale objects
The possibility of parts repair	Yes	No
Structural and compositional in situ modifications	Yes: easy to produce multi-structures and parts with compositional gradient, allows for in situ alloying	No: limited to one type of powder/cycle
Mesh structures	No	Yes
Post-processing requirements	Yes	Yes
Costs	High: high power laser sources and robots required	Lower: more compact machines, easy to implement in industry
Application in the biomedical field	Currently very low	Quite advanced: e.g., implemented in dental cabinets for the manufacturing of personalized dental prostheses

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
