# Peer review of "Prototype Orthopedic Bone Plates 3D Printed by Laser Melting Deposition"

_materials, 2019, doi:10.3390/ma12060906_

Round 1

Reviewer 1 Report

In this manuscript, the authors report the results obtained from printing  exemplary bone plates with directed energy deposition. Indeed, the research with this additive manufacturing technology is scarce, especially to print implants.  Therefore, the work could be a welcome addition to the existing literature on the topic. However, the manuscript looks more like a technical report than a scientific paper. There is very weak discussion of the results. The two questions raised in Discussion should be moved to Introduction to justify the choice of the fabrication technology for bone plates. The manuscript needs substantial revision. Alternatively,  the authors may consider it to be a contribution to a conference on additive manufacturing. Before (re)submission, the authors may consider the following suggestions.

Title: if the work concerns exemplary bone plates, using a general term “implants” should be avoided. The same goes to the manuscript body.

In Introduction, it will be more straightforward if a table is presented to compare SLM and LMD. The reasons for choosing LMD to fabricate bone plates should be stated. The main objective of the research should be specified.

In 2.1, the baseplate (substrate) temperature should be mentioned as well as the process parameters used for optimization. It will be nice to show the CAD design of the plates with dimensions.  Part of 3.4 should be moved to Materials and Methods.

In 2.3, EDS may not be the right technique to determine the composition of the powder bulk.  

In 3.1, characteristic powder particle sizes and size distribution (a graph from a laser sizer) should be reported instead of qualitative analysis of SEM images.

3.2 should be restructured to show the effects of the process parameters on the quality of printed plates (what are the quality indexes?), based on the research design in 2.1.

In 3.3, the measurements from the camera attached to the robotic arm may be presented to support the explanation of the formed phases due to high build chamber temperatures.

In 3.3, is the BCC Ti-Al-V the beta phase? Why is it disordered? How reliable was the determination of the mass concentration of the beta phase ? Fig. 7 shows the deviations of the as-printed composition from the powder composition. Why? Is the EDS technique suitable and what are its reliability and accuracy?

In 3.5, is elemental segregation detected to justify the in vitro tests (see the conclusion No. 2: excellent compositional uniformity)?

Author Response

In response to Reviewer 1’s comments

The authors are thankful for the careful review performed by Referee no. 1 which helped us improve the quality of the manuscript.

The modifications are highlighted in text with cyan color.

Comment [C] 1: However, the manuscript looks more like a technical report than a scientific paper. There is very weak discussion of the results.

Response [R] 1: We partially agree with the comment of the reviewer. While most of the LMD biomaterials literature is devoted 100% to the characterization of printed materials, our manuscript aimed to supplementary address the manufacturing part and its problematics. Although most of the articles published in the materials science field do not emphasize the technical part of a device/material manufacturing, providing such insights with transparency is in our opinion of significant importance for a further accelerated development of this specific branch of implant fabrication techniques. We are aware that the description of the engineering steps could sometimes appear similar to a technical report. Further efforts have been dedicated to strengthen and enrich the discussion of the results.

C2: The two questions raised in Discussion should be moved to Introduction to justify the choice of the fabrication technology for bone plates.

R2: The requested modification has been performed, and in the process, the last paragraph of the Introduction section has been improved to justify in a clearer way the goals of our research study: “In a step forward, we report and discuss on the LMD manufacturing conditions leading to fabrication of orthopedic bone plates. The main objective of the study was to produce and characterize Ti6Al4V shapes aimed for the future development of patient-customized 3D implants. The encountered technical difficulties that could impede this method to evolve from research stage to actual production of functional medical devices, will be underlined, and some possible technical solutions to overcome these shortcomings, will be highlighted. Answers to two highly relevant topical questions will be provided: “Why 3D printing?” and “Why so many technological steps need to be involved in the implant manufacturing by 3D printing?”. Besides the optimization and the technological steps of the manufacturing process, a thorough structural and compositional investigation of the LMD-grown bulk material is presented together with cytocompatibility studies, assessing its preliminary biological performance.” (page 3, lines 77-87)

C3: Title: if the work concerns exemplary bone plates, using a general term “implants” should be avoided. The same goes to the manuscript body.

R3: We changed “implants” by “bone plates” in both the title and the text, with the exception of general cases which referred specifically to actual implants or prostheses.

C4: In Introduction, it will be more straightforward if a table is presented to compare SLM and LMD. The reasons for choosing LMD to fabricate bone plates should be stated. The main objective of the research should be specified.

R4: As requested a table was added to the Introduction section, comparing the SLM and LMD techniques. We have introduced the reasons for selecting LMD as method of choice for bone plate manufacturing (page 2-3, lines 67 – 70). The main objective of the study has been now clearly expressed in the last paragraph of the Introduction section (page 3, line 78-79).

C5: In 2.1, the baseplate (substrate) temperature should be mentioned as well as the process parameters used for optimization. It will be nice to show the CAD design of the plates with dimensions. Part of 3.4 should be moved to Materials and Methods.

R5: The substrate was kept at room-temperature during printing. We have added now this information in section 2.1. The exemplary plate dimensions are presented in Fig. 14. The separation between the meander lines was specified in the text, since we preferred to emphasize the robot path in Figs. 10 and 12. Such a zooming effect did not allow for the concomitant presentation of both meander lines and dimension markings. We rather prefer to keep the dedicated chapter for bone plates manufacturing as is, since this represents a major result of our study, accompanied by representative figures of the fabrication steps. We hope that the reviewer will understand and accept our kind request.

C6: In 2.3, EDS may not be the right technique to determine the composition of the powder bulk.

R6: We agree with the reviewer that EDS is not the standard technique for determining the composition of alloys. However, unfortunately, a spark OES apparatus is not available in our Institutes. We have decided to study by EDS both the printed bulk and the source powder material, such as to obtain a comparative assessment of the compositional changes occurring during laser manufacturing. Please note, that the EDS software include a series of procedures, in agreement with the regulations of ISO 22309:2011 – “Microbeam analysis — Quantitative analysis using energy-dispersive spectrometry (EDS) for elements with an atomic number of 11 (Na) or above”, which states that EDS can be used with “confidence” for “quantitative analysis of mass fractions down to 1%” “for elements with atomic number Z >10”.  The technique shows good results even with critical specimens like powder samples.

C7: In 3.1, characteristic powder particle sizes and size distribution (a graph from a laser sizer) should be reported instead of qualitative analysis of SEM images.

R7: Unfortunately, our Institutes are not equipped with laser granulometer facility. Moreover, using the Mie scattering principle, we would have had difficulties in quantifying some subpopulations of particles (i.e., the small diameter particles soldered on the bigger diameter ones).

In order to comply with the reviewer’s request, we have added a new figure (Fig. 2c) with the particle size distributions extracted on the basis of SEM micrographs using the Image J software. The text has been also amended accordingly in both Materials & Methods (“The particle size distribution analysis has been performed on the basis of SEM micrographs (collected on randomly selected areas) with the help of Image J software (National Institutes of Health, USA).”) and Results (“Representative SEM images (Fig. 2a,b) of the Ti6Al4V powder, together with the particle size distribution (Fig. 2c). The powder particles are spherically-shaped (Fig. 2a). At higher magnification (Fig. 2b) small spherical particles soldered to the prominent larger particles, both with smooth morphologies were revealed. The particle size analysis, performed on the basis of SEM micrographs, revealed two particle populations: (i) one with diameters in the range 10 – 40 µm, best-approximated with a log-normal distribution having a median value of ~12 µm, and (ii) one with sizes in the range 50 – 130 µm, well-fitted by a Gaussian function, with a median value of ~71 µm.”) sections.

C8: 3.2 should be restructured to show the effects of the process parameters on the quality of printed plates (what are the quality indexes?), based on the research design in 2.1.

R8: The process parameters as shown in Fig. 3 were selected by laser-depositing Ti6Al4V lines on the titanium substrate. The line which had the best resolution (corresponding to a laser power of 1000 W and scanning speed of 0.3 m/min) was then used to manufacture the plates. We have not fabricated plates for all tested experimental conditions, using as quality factors the compromise between resolution and residual powder.

C9: In 3.3, the measurements from the camera attached to the robotic arm may be presented to support the explanation of the formed phases due to high build chamber temperatures.

R9: The camera attached to the robotic arm transmits in real time on a monitor. Its signal does not pass through a computer, and therefore images cannot be saved or processed. This camera serves only for monitoring the processing area for unexpected troubleshooting.

C10: In 3.3, is the BCC Ti-Al-V the beta phase? Why is it disordered? How reliable was the determination of the mass concentration of the beta phase?

R10: Yes, the bcc Ti-Al-V is the beta phase. The XRD analysis clearly revealed two crystal phases in the LMD Ti-Al-V material, a hexagonal one which is similar to that of the powder material, and a cubic one. The cubic Ti-Al-V phase is named in the literature β-phase in contrast to the hexagonal, which is named α-phase. The result agrees well with the microstructure revealed in the optical microscopy image presented in Fig. 5c. The “disorder” of the β-phase, evidenced by the large value of the microstrain, is probably related to the non-equilibrium nature of this phase at room-temperature, this phase being preserved at cooling due to mechanical constraints caused by α-phase expansion. The determination of the mass concentration of this phase by Rietveld quantitative analysis has not an excellent precision (but probably it is not worse than ±10%). We inserted in the manuscript the figures displaying the goodness of the fit of the XRD pattern, without a quantitative assessment of the determination errors (Fig. 7). Overall, the XRD results have been analyzed more insightfully and commented in more detail.

C11: Fig. 7 shows the deviations of the as-printed composition from the powder composition. Why? Is the EDS technique suitable and what are its reliability and accuracy?

R11: As mentioned in response to comment 6, EDS is considered a reliable and accurate technique for compositional quantitative analysis (i.e. mass fractions down to 1% for elements with atomic number Z >10). Statistically significant differences (as calculated by Student t’test method) have indicate compositional modifications only in the case of Ti, whose concentration dropped with ~0.9% in the case of the bone plate. This value is close to the resolution limit of EDS, according to ISO 22309:2011.

C12: In 3.5, is elemental segregation detected to justify the in vitro tests (see the conclusion No. 2: excellent compositional uniformity)?

R12: The in vitro tests have been performed to asses if the modification of crystalline phase composition, introduced during printing, are inducing or not cytotoxic effects with respect to a human (application relevant) cell line.

Reviewer 2 Report

A good article that provdes insight into the applicability of LMD Ti-6Al-4V for the medical sector. Ahead of publication the following points should be addressed:

Line 16 – powder debit? What does this present, powder feed rate, powder type? Please make this clear as the term powder debit is not used throughout the article.

Line 42 – disputing supremacy; reword ‘typically considered’

Line 60 – dimensions

Line 85 – Could you expand on this optimization step?

Line 102 – reword sentence from ‘and further on cut…’

Line 124 – First two sentence should be worded into one to describe the etchant/process

Line 136 – mm2

Line 186 – Could you provide more information on particle size distribution with the 2 different size ranges?

Fig 2 – improve clarity of plot 2c – axis size/labels

Line 219 – hours-long manufacturing irradiation durations

Line 227 – raising set-up 0.2mm for each line – this is better described as a build height between layers

Line 242 – this paragraph describes the typical LMD resulting microstructure from other articles, can you provide a quantified microstructure from you study and how you have performed the measurements?

Line 246 – more stable…

Line 269 – some of the symbols appear to have dropped out in this paragraph

Line 314 – Figure caption V.10

Line 437 – Check the powder sizes given in the conclusions match up to the body of text

Author Response

In response to Reviewer 2’s comments

The authors thank the reviewer for her/his precious help in increasing the manuscript’s overall quality. All improvement suggestions and typos have been adopted or corrected, as shown further.

The modifications suggested by reviewer 2 are marked in green.

Comment [C] 1: Line 16 – powder debit? What does this present, powder feed rate, powder type? Please make this clear as the term powder debit is not used throughout the article

Response [R] 1: We have modified “powder debit” by “powder feed rate”.

C2: Line 42 – disputing supremacy; reword ‘typically considered’

R2: We have changed “disputing supremacy” to “typically considered”.

C3: Line 60 – dimensions

R3: We have corrected the typo.

C4: Line 85 – Could you expand on this optimization step?

R4: We have added the following explanation at page 3, lines 97 – 102: “Over 3 g/min, the growth rate increase was not significant enough to justify for the high amount of consumed powder. Moreover, when surpassing 3 g/min of blown powder, besides the deposition of interest, an unwanted deposition of sputtered material occurred. This was caused by unmolten blown powder which adhered to the hot substrate in the vicinity of the laser irradiated area. Bellow 3 g/min, the quality of samples varied, from discontinuous deposition (up to 1 g/min) to porous and rough samples (up to 2.5 g/min).

C5: Line 102 – reword sentence from ‘and further on cut…’

R5: We rewrote the initial phrase as:First, an incipient shape was printed and further on sliced using a disk cutting machine, model Brillant 200 (ATM, Germany). An intermediate implant shape with thickness of 1 mm was thus obtained.” (page 4, lines 118 – 120)

C6: Line 124 – First two sentence should be worded into one to describe the etchant/process

R6: We combined the two phrases into one. This section of the manuscript now reads as: “In order to reveal the metallographic structure, the polished surfaces were chemically etched with a mix of HF (20%), HNO3 (10%) and water (70%).” (page 4, lines 144-145)

C7: Line 136 – mm2

R7: We have marked now the “2” as exponent. It is now “mm2”.

C8: Line 186 – Could you provide more information on particle size distribution with the 2 different size ranges?

R8: In order to comply with the reviewer’s request, we have added a new figure (Fig. 2c) with the particle size distributions extracted on the basis of SEM micrographs using the Image J software. The text has been also amended accordingly in both Materials & Methods (“The particle size distribution analysis has been performed on the basis of SEM micrographs (collected on randomly selected areas) with the help of Image J software (National Institutes of Health, USA).”) and Results (“Representative SEM images (Fig. 2a,b) of the Ti6Al4V powder, together with the particle size distribution (Fig. 2c). The powder particles are spherically-shaped (Fig. 2a). At higher magnification (Fig. 2b) small spherical particles soldered to the prominent larger particles, both with smooth morphologies were revealed. The particle size analysis, performed on the basis of SEM micrographs, revealed two particle populations: (i) one with diameters in the range 10 – 40 µm, best-approximated with a log-normal distribution having a median value of ~12 µm, and (ii) one with sizes in the range 50 – 130 µm, well-fitted by a Gaussian function, with a median value of ~71 µm.”) sections.

C9: Fig 2 – improve clarity of plot 2c – axis size/labels

R9: The clarity and resolution the indicated figure (now Fig. 2d) has been improved along with the Fig. 2a,b, such as to be easier to read.

C10: Line 219 – hours-long manufacturing irradiation durations

R10:hours-long irradiation durations” was replaced by “hours-long manufacturing irradiation durations”.

C11: Line 227 – raising set-up 0.2mm for each line – this is better described as a build height between layers

R11: We reworded as: “Each build layer was 0.2 mm thick and the nozzle was raised accordingly after each pass.

C12: Line 242 – this paragraph describes the typical LMD resulting microstructure from other articles, can you provide a quantified microstructure from you study and how you have performed the measurements?

R12: For the assessment of grain sizes we can provide only qualitative estimations, since their acicular shape and intertwining prevented a reliable quantitative microstructural analysis. Micrographs analysis software such as Image J and even SPIP did not deliver satisfying results. Moreover, due to the irregularity and superposition of these fine elongated grains, the manual large scale quantification was not possible. This is the reason why in the revised version of the manuscript we have decided to extend and delve more insightfully into the XRD data, to make possible quantitative assessment of the crystallite size, lattice parameters, micro-strain and phase composition.

C13: Line 246 – more stable…

R13: “stabler” was replaced by “more stable”.

C14: Line 269 – some of the symbols appear to have dropped out in this paragraph

R14: We have corrected the symbols mismatching. This was caused by incompatible Word variants between co-authors. We also checked the whole manuscript in order to eliminate all such occurrences.

C15: Line 314 – Figure caption V.10

R15: “10” was deleted. We thank the reviewer for noticing this parasite misspell.

C16: Line 437 – Check the powder sizes given in the conclusions match up to the body of text

R16: This error was corrected. The text is now written: “Orthopedic bone plates of Ti6Al4V with shape and dimensions similar to the commercial laser cut ones, starting from powder materials with two populations of spherical particles (with diameters in the ranges 10 – 40 µm and 50 – 130 µm) have been printed by LMD.” (page 17, lines 494 – 496)

Reviewer 3 Report

The paper deals with Ti6AL4V biomedical implants build by additive manufacturig and, in particular, by LMD. The paper is interesting and well written and I believe that the Readers could really apppreciate it. 

There are  afew typos that must be corrected in Lines 260-283, where symbols related to the Ti phases have been mistaken and should be corrected.

I would laso recommend to use the apex in the units of volume (Line s 262-263).

Author Response

In response to Reviewer 3’s comments

The authors thank the reviewer for her/his precious help in increasing the manuscript’s overall quality. All improvement suggestions and typos have been adopted or corrected, as shown further.

The modifications suggested by reviewer 3 are marked in yellow.

Comment [C] 1: The paper deals with Ti6Al4V biomedical implants build by additive manufacturig and, in particular, by LMD. The paper is interesting and well written and I believe that the Readers could really apppreciate it.

Response [R] 1: We thank the reviewer for her/his kind considerations about our manuscript.

C2: There are a few typos that must be corrected in Lines 260-283, where symbols related to the Ti phases have been mistaken and should be corrected.

R2: We have corrected the symbols mismatching. This was caused by incompatible Word variants between of co-authors. We also checked the whole manuscript in order to eliminate all such occurrences.

C3: I would also recommend to use the apex in the units of volume (Lines 262-263).

R3: We corrected and used the superscript for volume. It is now written everywhere as “Å3”.

Round 2

Reviewer 1 Report

The authors have properly addressed most of the issues that I raised. I understand the limited accessibility of the authors to equipment for certain characterization work and appreciate the intention of the authors to offer technical details at the trial stage of laser melting deposition (LMD) before bone plates were printed.  However, there are still some issues that the authors need to look at.

In 2.1, the Ti6Al4V bone plate were produced at room temperature. It is incorrect. If you mean the substrate temperature was maintained at room temperature, please say so, but you need to tell how it was achieved (through the cooling of the build plate?).

In 2.3, it is necessary to specify the number of EDS spot analyses of the powder and the solid structure, which allowed you to come up with the mean value, standard deviations and statistical comparison (Fig. 8).

In 2.3, the number of SME images used for the analysis of powder particle size distribution needs to be specified.

In 3.3, how was the mass concentration of the beta phase determined? Details must be given.

In 3.3, why was the difference in Ti concentration statistically significant while the differences in Al and V concentrations between the powder and solid structure were statistically insignificant? What happened to the alloy composition during laser melting deposition? Explanations are necessary.

In line 338, you say no visible elemental segregation was found, which contradicts with the sentence in line 415 (there is a possibility that elemental segregation might occur during LMD).

In 4, you explain the similar microstructures of the alloy produced by LMD and EBM due to the high build temperature. This explanation is not really convincing. In EBM,  the build plate temperature is as high as 650 – 700 degrees Celsius. However, in your work, the substrate temperature was kept at room temperature during printing  (your reply R5), which led to quick heat dissipation from the plate being built, instead of heat accumulation. Obviously, the thermal histories of the alloy during LMD and EDM are totally different. It is not clear in your case how in situ annealing heat treatment to cause the decomposition of the alfa prime phase would take place. Have you look at the microstructure variation along the z direction where the thermal history changes?  Please provide more convincing explanations.

Careful editing and corrections of grammatical errors are needed.

Author Response

We present hereunder the responses to the Reviewer’s comments, whilst resultant text additions/modifications have been performed in the manuscript (colored in red).

Comment [C] 1: In 2.1, the Ti6Al4V bone plates were produced at room temperature. It is incorrect. If you mean the substrate temperature was maintained at room temperature, please say so, but you need to tell how it was achieved (through the cooling of the build plate?).

Response [R] 1: This was a misunderstanding. There are some papers that start the LMD printing with heated substrates. We just wanted to specify that in our case, we used no additional heating method. During LMD of the bone plates, the Ti substrate heats up to 300 oC due to laser irradiation, as shown by a chromel-alumel thermocouple in contact with the sample during deposition. It is difficult to specify in the manuscript a substrate temperature during processing, as this is not constant and is dependent on the substrate size, laser power, spot size and irradiation time.

To avoid misunderstanding during reading, we corrected the text in the manuscript as: “The Ti substrates were not heated before LMD printing. However, the substrate temperature was monitored with a chromel-alumel thermocouple. In our experimental conditions, during bone plates printing, the substrate temperature reached a maximum of 300o C.” (Line 141)

C2: In 2.3, it is necessary to specify the number of EDS spot analyses of the powder and the solid structure, which allowed you to come up with the mean value, standard deviations and statistical comparison (Fig. 8).

R2: The EDS analyses were not performed on spots, but by scanning large areas (SEM fields) of the specimen to average over the possible uniformities. Four SEM fields (collected at a magnification of 500X, resulting into scan areas of 533 x 360 square micrometers) were randomly selected on the surface of the specimens, and the quantitative results were to infer the mean and the standard deviation. In the case of the source powder, the SEM specimen was prepared by pressing the powder against the carbon conductive adhesive tape in form a plane continuous coating. The number of analyses for each type of sample is now imprinted also on the new version of Fig. 8.

C3: In 2.3, the number of SME images used for the analysis of powder particle size distribution needs to be specified.

R3: Four SEM randomly collected images, having areas of 1340 x 905 square micrometers have been used to infer the particle size distribution in the case of the source powder. Close to 700 individual particles sizes have been measured and included for the powder size distribution determinations. The SEM specimen was prepared by sprinkling powder on the surface of the carbon conductive adhesive tape to allow for the facile visualization of individual particles.

C4: In 3.3, how was the mass concentration of the beta phase determined? Details must be given.

R4: The required information concerning the procedure is now inserted in section 3.3 (please see the last paragraph on page 9, line 433).

C5: In 3.3, why was the difference in Ti concentration statistically significant while the differences in Al and V concentrations between the powder and solid structure were statistically insignificant? What happened to the alloy composition during laser melting deposition? Explanations are necessary.

R5: For the unpaired Student’s t-test we used, a probability value of 0.05 (95% confidence) was considered to determine if the recorded compositional differences are statistical significant or not. A p-value < 0.05 would provide evidences against the null hypothesis, allowing for its rejection; a p-value > 0.05 would deliver feeble evidences against the null hypothesis, consequently resulting in the failure of rejecting the null hypothesis; a p-value close to 0.05 is typically interpreted as marginal.

As in the case of Ti concentration, the determined p-value (i.e., 0.03748, mentioned in the original version of the manuscript) could be considered by some to close to the cut-off, we decided to presented it as such, allowing for the readers to judge uninfluenced the result. During this second revision we have inserted also the p-values calculated for Al and V.

Furthermore, even if interpreted as statistical significant, the 0.92 wt.% difference in Ti concentration between source powder and LMD sample is situated (as mentioned in the manuscript revised in round 1) very close to the EDS quantitative resolution limit of 1 wt. % for elements with atomic number Z >10.

In such conditions, advancing all sort of hypotheses concerning the Ti concentration differences would be in our opinion a bit far-fetched, as we would rely on frail foundations (as shown above).

C6: In line 338, you say no visible elemental segregation was found, which contradicts with the sentence in line 415 (there is a possibility that elemental segregation might occur during LMD). R6: No, there is no visible elemental segregation or depletion in EDXS maps. We deleted this statement from the in-vitro chapter, in order to avoid confusion.

C7: In 4, you explain the similar microstructures of the alloy produced by LMD and EBM due to the high build temperature. This explanation is not really convincing. In EBM, the build plate temperature is as high as 650 – 700 degrees Celsius. However, in your work, the substrate temperature was kept at room temperature during printing (your reply R5), which led to quick heat dissipation from the plate being built, instead of heat accumulation. Obviously, the thermal histories of the alloy during LMD and EDM are totally different. It is not clear in your case how in situ annealing heat treatment to cause the decomposition of the alfa prime phase would take place. Have you look at the microstructure variation along the z direction where the thermal history changes?  Please provide more convincing explanations.

R7: In light of our answer at C1, it becomes clear that LMD and EBM can be similar in terms of microstructures. The build plate temperature reaches hundreds of degrees Celsius in both cases, while in the irradiation area it surpasses the melting temperature of Ti. Cooling is not fast, as the printed incipient shape is continuously irradiated by the laser beam at a very slow speed. Ref 39 shows a typical α+β structure obtained by EBM which is similar to ours, while also presenting a typical laser melting structure which is α’ martensitic and which looks quite different.

We have not performed a shot by shot investigation along the z direction, but searched along the whole area of the sample cross-section for clear microstructure differences. We have not identified such areas. Everywhere, there are former β grains in which α phase grew in various proportions. There are isolated grains in which martensitic α’ orthogonal grains can be identified. Due to limitations of this answering field, in the Word version of this response, two suggestive images for the whole cross-section of a LMD printed bone plate are presented.

C8: Careful editing and corrections of grammatical errors are needed.

R8: The manuscript has been read again by the authors and numerous semantic modifications have been performed in order to increase the manuscript’s clarity and English level. Please check the new version of the manuscript with track changes for English revision.
